# The Scarcity of Specific Nutrients in Wild Bee Larval Food Negatively Influences Certain Life History Traits

**DOI:** 10.3390/biology9120462

**Published:** 2020-12-11

**Authors:** Zuzanna M. Filipiak, Michał Filipiak

**Affiliations:** Faculty of Biology, Institute of Environmental Sciences, Jagiellonian University, Gronostajowa 7, 30-387 Kraków, Poland

**Keywords:** nutritional stress, nutrient cycling, ecological stoichiometry, ecosystem, bee, sodium, potassium, zinc, food, diet

## Abstract

**Simple Summary:**

Pollen comprises many organic substances (sugars, lipids, proteins, amino acids, vitamins, etc.), all of which are built from elements such as carbon, hydrogen, oxygen, nitrogen, phosphorus, sodium, potassium, zinc, and approximately twenty others. These special nutritional elements compose the cells, tissues, and bodies of all the life forms on our planet and are needed by bee larvae for healthy growth. However, not all plants produce pollen containing these elements in proportions needed specifically by bees, meaning that not all pollens are nutritionally balanced for bees. Moreover, the decrease in plant diversity is thought to be among the main causes of the dwindling numbers of pollinators worldwide. Currently, governments and societies are attempting to combat this pollinator decline by providing nutritionally balanced and diverse food plants to pollinators. Knowing which nutritional elements are crucial for the bee diet and understanding why are prerequisites for tailoring conservation efforts for this group of insects, which are substantially important for human nutrition and ecosystem functioning. Basic information obtained from feeding experiments is important for synergistically understanding how plant diversity within certain species that produce pollens with rich or scarce amounts of certain nutritional elements influences bee health and prosperity.

**Abstract:**

Bee nutrition studies have focused on food quantity rather than quality, and on details of bee biology rather than on the functioning of bees in ecosystems. Ecological stoichiometry has been proposed for studies on bee nutritional ecology as an ecosystem-oriented approach complementary to traditional approaches. It uses atomic ratios of chemical elements in foods and organisms as metrics to ask ecological questions. However, information is needed on the fitness effects of nutritional mismatches between bee demand and the supply of specific elements in food. We performed the first laboratory feeding experiment on the wild bee *Osmia bicornis*, investigating the impact of Na, K, and Zn scarcity in larval food on fitness-related life history traits (mortality, cocoon development, and imago body mass). We showed that bee fitness is shaped by chemical element availability in larval food; this effect may be sex-specific, where Na might influence female body mass, while Zn influences male mortality and body mass, and the trade-off between K allocation in cocoons and adults may influence cocoon and body development. These results elucidate the nutritional mechanisms underlying the nutritional ecology, behavioral ecology, and population functioning of bees within the context of nutrient cycling in the food web.

## 1. Introduction

Diminished floral nutritional resources are among the most important factors causing declines in bee species richness and abundance [1,2], and knowledge about their nutrition is needed for accurate and targeted measures for the protection and restoration of bee species [3,4]. Most information concerning the nutrition of bees comes from studies on managed honeybees (*Apis mellifera*) [5,6] and wild bumblebees (e.g., *Bombus terrestris*) ([7] and references therein). However, their life history traits are different from those of solitary bees, representing more than 75% of all bees on Earth [8]. One main difference between eusocial (living in colonies, e.g., the western honeybee) and solitary bees is associated with the provision of food to progeny; for the former, food is steadily provided to progeny, whereas females of solitary bees supply their offspring only once. The quantity and, most importantly, the quality of pollen is of great importance for the survival, development, and maturation of solitary bees, and thus for individual fitness and population persistence [7,9]. Information about the nutritional demand of solitary bees and the relationships among larval food quality, adult fitness and population prosperity is limited [1,7,10]; therefore, novel approaches for studying this topic are required [2,4,10].

To date, studies considering the nutritional quality of bee food have focused mainly on the total contents or ratios of proteins, lipids, and carbohydrates (e.g., [11,12,13]); individual amino acids [5,14]; species composition and the proportions of different species in pollen diets [15,16]; and bee cognition and behaviors related to the preference for particular plant species [17,18,19]. The elucidation of biochemical and behavioral properties greatly enhances our knowledge of the feeding behaviors of bees, and knowledge about such organism-level phenomena is important for bee conservation efforts. However, a broader context is needed for understanding ecosystem functioning, where not only the biology of individual bees but also the functioning of bees in ecosystems and their involvement in complicated sets of ecological interactions are considered [20]. Taking this into account, ecological stoichiometry was proposed as a complementary viewpoint on wild bee nutrition [9,10] that not only allows us to study the nutritional preferences of individuals but also considers the organism as part of the ecosystem [21].

According to the ecological stoichiometry framework, nutritional supply is available in the environment in the form of foods composed of organic molecules, and every molecule is composed of a specific set of atoms of chemical elements, which are utilized by organisms [20]. The immutable atoms cycle in food webs in endless loops and are ceaselessly incorporated into molecules that are constantly degraded through the biogeochemical cycle [22,23,24]. Therefore, we herein adopt the biochemistry-oriented approach in which digested molecules are processed and the immutable atoms that they comprise are allocated to specific functions playing specific roles in the fitness and life histories of organisms. The element-based approach is considered evolutionary [25] and ecologically relevant [26], especially for soil–plant–pollinator interactions under anthropogenic changes where the input (e.g., fertilization) or removal (e.g., harvest) of elements to/from the environment can be observed. To understand the roles of atoms of chemical elements in the biology and ecology of the considered organisms and relate the concentrations of specific elements in food to specific life history traits that shape bee fitness, basic knowledge about the nutritional values of those elements for organisms is needed.

The goal of this study was to determine the impact of Na, K, and Zn deficiencies on the fitness and life history traits of the generalist solitary bee *Osmia bicornis* L. In their review, Filipiak and Weiner [9] indicated that the growth and development of *O. bicornis* may be limited by the scarcity of K, Na, and N in pollen, whereas cocoon development may be limited by P, Mg, K, Na, Zn, Ca, and N deficiencies. The importance of Na (and K via Na:K ratio) for the functioning of ecosystems in general was emphasized by Kaspari [27]. Moreover, the scarcity of K, Na, and Zn in *O. bicornis* larval food was indicated as being potentially limiting for bee growth and development [28]. Both K and Na are essential for life, with a gradient generated by Na-K-ATPase, which is critical for the maintenance of osmotic balance and the resting membrane potential of most tissues as well as for muscle and nerve cell excitability [29,30]. Zinc, in turn, is a catalytic component of more than 300 enzymes and a structural component of many proteins and enzymes [31,32].

Using a feeding experiment, we determined the fitness-related life history traits (mortality, developmental cocoon stage, and dry mass of the developed adult body) of *O. bicornis* after exposure to control and nutrient-deficient pollen from the juvenile (three-day larvae) to the adult (imago) stage. In addition, individuals were exposed to pollen supplemented with K, Na, or Zn to confirm or exclude the effects of scarcity of a given element.

## 2. Materials and Methods

### 2.1. Model Organism

*Osmia bicornis* (*O. rufa*, Hymenoptera: Megachilidae) wild bees were obtained from a nest trap assembled with ca. 500 empty *Phragmites* sp. stems (250–300 mm in length; 6–10 mm in diameter) in the form of a case. The nest was located in the vicinity of the Institute of Environmental Sciences, Jagiellonian University, Kraków, Poland (50°01′35″ N; 19°54′05″ E). Female *O. bicornis* constructed their nests in the cane stems.

The nesting biology of the bee is shown in Figure 1 and was previously described in detail by Filipiak [10]. Usually, female eggs are laid first; therefore, they are located in the rear part of the nest, whereas male eggs can be found near the entrance [8,33]. In early spring, once females started to construct their nests, the stems were checked daily for the presence of closed brood cells. Firstly, stems (N = ca. 250) with ca. 1–3 closed brood cells were collected to obtain female larvae, and when the bees closed the tubes with mud, more stems (N = ca. 250) were collected to obtain male larvae.

All stems were kept at 21 °C and 60% relative humidity (RH) under a 12:12 (L:D)-h photoperiod. Only specimens from the first (females) and last (males) brood cells within each stem were collected for the experiment. Due to the fragility and sensitivity of eggs and possible mechanical damage to the eggs during transfer to experimental containers, 3-day-old larvae were used for the experiment.

### 2.2. Experimental Design

A feeding experiment was designed to determine fitness-related life history traits (mortality, developmental stage of the cocoon, and dry mass of the developed adult body) of solitary bee (*O. bicornis*) larvae fed food (pollen) nutritionally balanced or scarce in specific nutrients (physiologically important chemical elements: K, Na, and Zn). Fitness-related life history traits were chosen for study because: (1) mortality is an evident and relevant trait; (2) cocoons are fitness-enhancing secretions that protect bees for approximately ten months of pre- and overwintering [34,35]; and (3) body mass is positively related to fitness in *O. bicornis* and other solitary bee females but not males [36,37]. The three studied life history traits are considered separate and competing “sinks” into which organisms allocate resources from the available pool (see, e.g., [38] for more information). For ecological relevance and to make our experimental results relatable to the natural world, we analyzed and discussed the data obtained, focusing on the relevance of the studied traits for bee fitness.

Fifteen replicates (Eppendorf tubes, 2 mL) were prepared per treatment and sex and filled with homogenized pollen of specific nutritional quality. The amount of pollen corresponded to the dry masses of pollen provisions found in nature, i.e., 195 ± 5 mg dm for females and 140 ± 5 mg dm for males. Dry pollen loads were complemented with either demineralized water or salt solutions (to reach concentrations of the studied elements found in *Osmia*-collected pollen) in an amount reflecting ca. 25% of the dry pollen mass. Before starting the experiment, the Eppendorf tubes were left for 24 h to allow the water and salt solutions (KCl, NaCl, and ZnCl_2_) to penetrate the pollen loads. The three-day-old larvae were assigned to treatments with one individual per Eppendorf tube. All experimental tubes were kept at 21 °C and 60% RH under a 12:12 (L:D)-h photoperiod for 3 months. The exposure period was long enough to ensure that all larvae had reached the adulthood stage of the life cycle, i.e., the stage where fully developed individuals hibernated in their cocoons [8,35]. At the end of the exposure periods, cocoons were collected to determine the degree of cocoon development (in the case of undeveloped individuals who died as larvae and did not reach maturity, cocoons were not available). Then, the bees were extracted from cocoons, and the mortality rate was assessed. Afterwards, the individuals and cocoons were dried using a vacuum dryer (80 °C, 48 h) to obtain the dry mass.

### 2.3. Pollen Diets

Polyfloral pollen mixtures characterized by differing nutritional qualities expressed using the concentrations of the studied elements were used for the feeding experiment. Pollen mixtures were obtained from either *O. bicornis* provisions collected manually from brood cells or derived from commercially available polyfloral pollen pellets collected by honeybees (*Apis mellifera*) in central Europe. The pollen collected by *O. bicornis* in the field was considered a balanced diet, providing the needed amounts and proportions of nutrients to the bee larvae during development, and was used as a control diet in the experiment (Control-Osmia as described below). For practical reasons, only honeybee-collected pollen pellets could be used as diets depleted of certain nutrients in the experiments; therefore, we used an additional control diet (Control-Apis as described below), i.e., a diet similar to Control-Osmia in nutritional quality but composed of honeybee-collected pollen pellets.

Five packs of honeybee pollen pellets were purchased from different manufacturers and were composed of pollen of various botanical origins. According to the method proposed by Filipiak and colleagues [39], pollen from each pack was divided according to color by the naked eye to obtain pollen pellet pools with specific elemental compositions. Additionally, unsorted pools of pollen pellets were retained from each pack. In total, we obtained 15 different pollen pools from all packs: 5 unsorted and 10 sorted pools.

The concentrations of K, Na, and Zn were determined in all purchased pollen pools (unsorted and sorted) and in *O. bicornis* provisions. From all pollen pools, we chose the following pools for use in the feeding experiment (their nutritional qualities are given in the “Results” section): (1) control pollen from *O. bicornis* provisions, i.e., natural larval food, designated Control-Osmia; (2) control honeybee pollen, i.e., the unsorted pollen pellets obtained from one of the packs, which had a chemical composition similar to that of the *O. bicornis* provisions, designated Control-Apis; (3) three sorted honeybee pollen pellets pools with the lowest concentration of one of the studied elements (Na, K, or Zn), designated Na-deficient, K-deficient, and Zn-deficient; and (4) the same three sorted honeybee pollen pellets pools with the lowest concentration of one of the studied elements (Na, K, or Zn) and supplemented with salt containing the deficient element to reach the same concentration found in Control-Osmia, designated Na + supplemented, K + supplemented, and Zn + supplemented. The pollen pools from each treatment were homogenized manually to obtain a homogenous powder and then freeze dried to obtain the dry mass (dm).

### 2.4. Chemical Analysis

To analyze the K, Na, and Zn concentrations, freeze-dried pollen homogenates (five per treatment) were digested on a hotplate in a 4:1 mixture of nitric acid (70%) and hydrogen peroxide (30%). The K, Na, and Zn concentrations were measured using atomic absorption spectrometry (PerkinElmer AAnalyst 200 and PerkinElmer AAnalyst 800) and expressed in ppm dm. To determine the analytical precision, certified reference materials (bush, NCS DC 73349; chicken, NCS ZC 73016; and bovine muscle powder, RM8415) were examined with the samples.

### 2.5. Data Handling and Statistical Analysis

Differences in mortality between the nutritionally deficient and supplemented groups for each element separately, i.e., K-deficient vs. K + supplemented, Na-deficient vs. Na + supplemented, and Zn-deficient vs. Zn + supplemented, as well as between Control-Apis and the other groups were assessed using the Chi-squared test, with Yates’s correction for one degree of freedom.

The distribution of body mass was checked for normality with Shapiro–Wilk’s W test, and the homogeneity of variances was checked with Levene’s test. If the criteria were not met, the data were either log- or square root-transformed, and if these steps failed, a nonparametric (Kruskal–Wallis) test was used. Only individuals who survived until the end of the experiment, i.e., those that had undergone the entire metamorphosis to the imago form, were considered in the analyses of body mass and cocoons. The effects of the treatments on body mass were tested using the Kruskal–Wallis test.

For adult (imago) bees, the degree of cocoon development was assessed by qualitative analysis, and two stages of cocoon development were distinguished. (1) The first stage was an underdeveloped cocoon that covered only part of the bee or not at all, and the cocoon consisted almost exclusively of soft (“wooly”) fragments; the cocoon tore easily with bare hands but was impossible to cut using a knife because it was too soft. (2) The second stage was an almost fully or fully developed cocoon that covered the whole bee body and mainly consisted of a hard material; the cocoon was difficult or impossible to tear by bare hands but was possible to cut using a knife because it was sufficiently hard (Figure 2). The second stage of cocoon development might have the greatest probability of allowing an adult individual to overwinter until the next season and protecting the bee from external factors (e.g., parasites or pathogens). The differences in reaching the second cocoon developmental stage between the deficient and supplemented treatments (each element separately), and between Control-Apis and the other groups were assessed using the Chi-squared test, with Yates’s correction for one degree of freedom. All analyses were performed separately for females and males.

To calculate the percentages of each type of cocoon developed by the bees we considered all 15 bee specimens as 100%, constituting every treatment and control. Therefore, this 100% consisted of the sum of (1) specimens that reached the adult stage and developed to the first cocoon stage, (2) specimens that reached the adult stage and developed to the second cocoon stage and (3) specimens that died before reaching the adult stage (the majority of which did not reach the last larval stage, i.e., spinning larvae that start to produce the cocoon). Organismal death before reaching maturity has obvious negative consequences for fitness, therefore we separated these specimens in our analysis from those that successfully reached maturity and developed cocoons to make our analysis ecologically relevant. In this way, we treated the cocoon developmental stage as an ecologically meaningful trait that influences the fitness of living and mature bees.

This analysis was complemented by a simultaneous redundancy analysis (RDA) of the datasets on body mass and cocoon stages performed in Canoco 5 [40], which helped us to determine whether the negative effects of nutrient scarcity on these two life history traits (1) were correlated and (2) differed in strength.

## 3. Results

### 3.1. Pollen

The concentrations of the studied elements in the control and deficient pollen are presented in Table 1. The potassium concentrations in the Control-Apis, Na-deficient and Zn-deficient pollen loads corresponded to 97–101% of that in Control-Osmia, whereas the K concentration in the K-deficient treatment was ca. 26% lower than that in both control treatments. Sodium concentrations were ca. 1–16% higher in the Control-Apis, K-deficient, and Zn-deficient treatments and 39% lower in the Na-deficient treatment than in Control-Osmia. The zinc concentration in the Zn-deficient treatment was 39% lower than that in Control-Osmia, while the Zn concentrations in the Control-Apis, K-deficient, and Na-deficient treatments were ca. 2–7% lower than that in Control-Osmia.

### 3.2. Mortality

The highest mortality among females was observed in the Na-deficient treatment (80%), while the lowest mortality was observed in the Control-Apis and Zn + supplemented treatments (7%) (Table 2). For males, the highest mortality was observed in both Na treatments (73%) (Na-deficient and Na + supplemented), while the lowest mortality was observed in the Zn + supplemented treatment (0%). Significantly lower female mortality was observed for Control-Apis than for the K-deficient (*χ*^2^ = 4.26; *p* = 0.04), Na-deficient (*χ*^2^ = 13.57; *p* = 0.0002), and Na + supplemented (*χ*^2^ = 9.19; *p* = 0.002) treatments. For the other treatments, no differences in mortality were observed in comparison with Control-Apis (*p* > 0.3). Similarly, the mortality rates of male individuals in the Control-Apis treatment were significantly lower than those of male individuals in the K-deficient (*χ*^2^ = 7.35; *p* = 0.0007), Na-deficient (*χ*^2^ = 11.25; *p* = 0.0008), and Na + supplemented (*χ*^2^ = 11.25; *p* = 0.0008) treatments. In addition, significantly lower mortality was observed in the Zn + supplemented treatment compared with the Zn-deficient treatment (*χ*^2^ = 5.21; *p* = 0.02), with a nearly significant difference between Control-Apis and Zn-deficient (*χ*^2^ = 2.98; *p* = 0.08) and no difference between the Control-Apis and Zn + supplemented treatments (*χ*^2^ = 0; *p* = 1.0).

### 3.3. Cocoon Development

The percentages of each type of cocoon developed by bees are presented in Figure 3. In general, 73–74% and 66–73% of female and male bees, respectively, developed almost fully or fully formed cocoons (second stage) when reared on Control-Apis and Control-Osmia pollen.

The comparisons of cocoon status at the second developmental stage revealed that, for females, significantly fewer fully developed cocoons were observed in the K-deficient (*χ*^2^ = 11.25; *p* = 0.0008), Na-deficient (*χ*^2^ = 14.35; *p* = 0.0002) and Na + supplemented (*χ*^2^ = 4.80; *p* = 0.03) treatments than in Control-Apis. Similarly, significantly fewer fully developed male cocoons were observed in the K-deficient (*χ*^2^ = 12.15; *p* = 0.0005), Na-deficient (*χ*^2^ = 6.80; *p* = 0.009) and Na + supplemented (*χ*^2^ = 6.80; *p* = 0.009) treatments than in Control-Apis. In addition, significantly more developed cocoons were observed in the K + supplemented treatment than in the K-deficient treatment for both females (*χ*^2^ = 5.71; *p* = 0.02) and males (*χ*^2^ = 6.71; *p* = 0.01).

### 3.4. Imago Body Mass

The effect of treatment on body mass was significant for both females (*p* = 0.00002) and males (*p* ≤ 0.0001). In total, 78 females were included in the analysis, for which significantly lower body masses were observed for individuals in the K-deficiency, K + supplemented, and Na-deficiency treatments compared with Control-Osmia, and no differences were observed between the other treatments (Figure 4). For males, 75 individuals were included in the analysis. The body masses of male individuals in both the K-deficient and K + supplemented treatments were lower than those in the Control-Osmia and Zn + supplemented treatments. Moreover, significantly higher body masses were observed for males exposed to Zn + supplemented pollen than for males exposed to Zn-deficient pollen.

### 3.5. Simultaneous Redundancy Analysis (Body Mass Plus Cocoon Stage)

The RDA of the imago body mass and cocoon stage (Figure 5) suggested that the negative effects of nutrient scarcity in the diet on these two traits were not correlated. For females and males, the first two axes explained 36.82% and 45.36% of the total variance, respectively. Relationships between adult mass/cocoon stage and the experimental diets are denoted by vectors. For both sexes, a vector symbolizing the cocoon stage positioned between the axes was situated perpendicular to the vector symbolizing adult mass, with the number of well-developed cocoons in a treatment increasing from the right-lower to the left-upper corner of the graphs and the adult mass increasing contrarily from the right-upper to left-lower corner of the graphs. For both sexes, the vector symbolizing adult mass was larger than the vector symbolizing the cocoon stage, suggesting a stronger effect of nutrient scarcity on mass than on cocoon development. Similar to previous analyses, the strongest negative effect of nutrient scarcity on cocoon development was observed for K in both sexes; additionally, the RDA suggested a similar effect for Na in females. For males, a positive effect of Zn supplementation on body mass was revealed, similar to previous analyses.

## 4. Discussion

The comparative experimental approach presented in this study provides evidence that deficiencies in specific elements in larval food impose constraints on certain life history traits and on the fitness of wild bee *Osmia bicornis.*

An important but understudied component of bee nutritional ecology is the relationship between K and Na concentrations in food [27]. In our study, K deficiency had similar effects on both sexes as follows: reduced survivability, reduced body mass, and underdevelopment of cocoons. K supplementation improved survivability and increased the proportion of well-developed cocoons, but had no effect on body mass. This effect is in line with that suggested by a theoretical study demonstrating that the trade-off for K may occur between allocation to the adult bee body and allocation to its cocoon [9]. Such a phenomenon was observed in our study, where the allocation of K to cocoons resulted in a smaller body size.

In the current study, Na scarcity strongly reduced survivability for both sexes; however, Na supplementation had a slight positive effect on female fitness, which manifested as increases in adult body mass. Among female larvae fed Na-deficient and Na + supplemented pollen, only three and five individuals, respectively, survived, whereas for male larvae, four individuals survived on both Na-deficient and Na + supplemented pollen. The facts that only a small number of specimens survived and even fewer of them developed cocoons suggest that something other than sodium might have affected the bees. A possible explanation is the scarcity of other colimiting nutrients (apart from Na) or the presence of poisonous substances; for example, bees may be negatively affected if their food consists of a large proportion of pollen having unfavorable chemical properties [41]. Additionally, the digestibility of pollen from specific species might affect bee fitness [42]. The pollen pools used in our experiment were hand-sorted based on color, therefore species composition was not assessed for the pollen pools, and no nutrients other than K, Na, and Zn were analyzed in the pools. Thus, we cannot conclude with 100% certainty that Na scarcity was the sole driver of such low survival and poor development in bees fed Na-deficient pollen. Nevertheless, a slight but significant effect of Na supplementation on female body mass was observed irrespective of any factors that might have affected the outcome of our study. Importantly, body size positively influences the fitness of females but not males [36,37]; therefore, the observed effect has ecological relevance.

The levels of potassium and sodium are essential for homeostasis in living cells. Both elements are maintained in gradients that are involved in the maintenance of transmembrane electrochemical potential differences, which are essential for cell signaling and secondary transport [29]. Conversely, disruption of potassium and sodium cation gradients can result in paralysis or death. Regarding potassium, its homeostasis in insects is associated with adaptation to extreme cold and heat [30]. For example, studies on adults of the true bug *Pyrrhocoris apterus* and the beetle *Alphitobius diaperinus* revealed that during a seven-day exposure at low temperatures (−5 °C and 4 °C, respectively), a gradual increase in potassium cations was observed within the hemolymph of both species, whereas for the other studied elements, i.e., magnesium and sodium, almost no changes were observed, indicating the importance of potassium homeostasis in response to cold stress [43]. Overall, it is not surprising that potassium had such a strong influence on the survival and development of cocoons in our experiment. However, because plant tissues in general contain high levels of K and low levels of Na, the K:Na ratio in herbivores’ foods strongly influences their fitness and must therefore be adequately balanced [27]. For example, acute bee paralysis may be caused by an excessively high K:Na ratio in their food [27,44]. Although, in diverse floras, K is not expected to have a limiting effect on herbivorous insects, including wild bees; nonetheless, low levels of potassium can be found in the pollens of several plant species [9]. Therefore, in the case of monocultures or habitats with low species richness, such a phenomenon might occur. For instance, based on data available from the literature, stoichiometric mismatches were calculated for different pollen species, showing that *Silybum marianum, Olea europaea,* and *Lavandula* sp. produce stoichiometrically unbalanced pollen for *O. bicornis* bees in terms of the potassium content [28]. Importantly, these plants are usually grown in large agricultural areas.

Regarding sodium, its gradient maintains the secondary transport system, which mediates the transport of other ions, substrates (e.g., glucose), and neurotransmitters across the plasma membrane [29]. Most importantly, Na is one of the most limiting elements for herbivores [27], including bees [39,45], and strong preferences of different bee species for sodium have been shown [27,45,46,47]. The sodium concentration in pollen depends on the species and is the most variable among all the elemental concentrations studied, differing fivefold between the pollens of species with the minimum and maximum concentrations [9]. Therefore, considering the availability of Na for developing bees, the occurrence of plant species producing Na-rich pollen in bee habitats may be important for both females and males, potentially influencing the growth of the entire bee population.

Females and males differed in their responses to Zn levels. Supplementation with Zn had the strongest effect on males, with lower mortality rates and higher body masses being observed upon exposure to Zn + supplemented pollen in comparison with Zn-deficient pollen. Although the percentage of Zn in the body is estimated to not exceed 0.02%, Zn is the most important trace element for the proper functioning of various tissues, organs and systems [31]. For instance, Beanland and colleagues [48] showed that the proportion of Zn in relation to two other minerals (Fe, B) in soybean (*Glycine max*) affected the development of three herbivorous insects (*Pseudoplusia includens*, *Epilachna varivestis*, and *Anticarsia gemmatalis*). A study on Zn supplementation in the sucrose diet of the honeybee *A. mellifera ligustica* revealed that 30 mg Zn kg^-1^ in food was sufficient to maintain the antioxidative (Cu/Zn-SOD activity) status of bees and to increase the survival of worker bees in comparison to those of bees exposed to lower (0–15 mg Zn kg^-1^) and higher (>45 mg Zn kg^-1^) Zn levels [49].

Interestingly, it has been shown for various bees that adults of different sexes use different plant species as food resources [50]. In *O. bicornis,* larval diets composed of pollen gathered by a female for her daughters and sons differed in nutritional quality, and this difference reflected sex-specific nutritional optima [28]. Moreover, female *O. bicornis* bees have a higher demand for Zn than males [28]. In general, in our study, zinc was the diet element to which females were the least sensitive (in terms of mortality, cocoon development and body mass). The explanation for these results might be associated with the function of zinc in female bee reproduction. For example, Cane [51] showed that after emergence from the cocoon, adult female *Osmia californica* bees require access to pollen to mature their oocytes and reproduce. Wasielewski and colleagues [52] observed that the first oocytes and ovaries of *O. rufa* (*bicornis*) bees developed gradually during wintering, and the authors linked the development to the vitellogenin content, whereas Lee and colleagues [53] found that after diapause, the length of the ovary and first oocytes as well as the number of oocytes were correlated with the vitellogenin secretion level in *Osmia cornifrons*. Vitellogenic proteins are female-specific egg-yolk precursors transferred to oocytes, where they provide nourishment for embryos [54]. Interestingly, the vitellogenin content was found to be closely related to Zn levels in female honeybees, because this protein acts as a Zn carrier [55,56]. Thus, we hypothesize that bee mothers provide both female and male eggs with pollen that contains a sufficient Zn level for development and functioning, but at later stages (i.e., after emergence from the nest), females can replenish zinc levels for continued functioning, e.g., vitellogenin and egg production, by eating pollen [51]. Importantly, similar supplementation strategies for other nutritional elements are impossible because adequate amounts and ratios of these elements are needed during larval growth and pupation. Therefore, Zn deficiency might exert constraints on developing males, manifesting as reduced survivability and body mass, whereas in females, Zn scarcity during the larval stage might negatively affect the reproductive system (not studied in our experiment). The reproductive system might be further rebuilt by adult females to ameliorate this negative effect.

Various bees, even those feeding on a variety of plant species, show preferences for particular plant species as food sources, especially considering pollen food for larvae [3,57,58]. Moreover, these preferences may be driven by specific nutritional needs reflected in the chemistries of the gathered food [13,59,60,61]. However, the biochemical metrics commonly used in bee nutrition studies, although ideal when focusing on bee biology, seem to be insufficient when considering the bee as part of the ecosystem and biogeochemical cycle—an organism involved in nutrient cycling. Therefore, adopting approaches complementary to traditional approaches, such as the biochemistry-oriented view, and focusing on nutrient flow through ecosystems allow for a better understanding of interactions between pollinators and other food web components (e.g., soil–plant–pollinator interactions) [62,63,64]. According to Paseka and colleagues [26], the frequency of element colimitation in terrestrial ecosystems suggests that stoichiometric effects on plant productivity may, in turn, affect pollen production and thus pollinators, although no studies on the relationships between elemental ratios in the environment and pollen production have been performed.

## 5. Conclusions

Bee conservation efforts are often based on simplistic assumptions, considering the nutritional ecology of only one life stage (usually adults) or sex (usually females). However, bee populations consist of individuals of various life stages and different sexes. Effective management strategies for maintaining populations of wild bees may be achieved only by obtaining and understanding the relationships between the complex nutritional demands of the whole bee population and the nutritional supply of pollen produced by different plants, including sex and life-stage differences in bee nutritional needs. Within this context, the current study provides the first insight into the effects of specific-atom scarcity in larval food on the life history traits and fitness of bees, thereby revealing the nutritional mechanisms underlying the nutritional ecology, behavioral ecology and population functioning of bees within an ecosystem context.

In this study, we confirmed earlier theoretical predictions, showing the following:*O. bicornis* life history traits and fitness are shaped by the availability of atoms of specific chemical elements in larval food.Some of these traits might be shaped by the availability of specific elements in a sex-specific manner: Na might influence female body mass, whereas Zn might influence the mortality and body mass of males.A trade-off between the K allocation to cocoons and the adult body may exist and might influence the development of cocoons and the body mass of adult bees.

## Figures and Tables

**Figure 1 biology-09-00462-f001:**
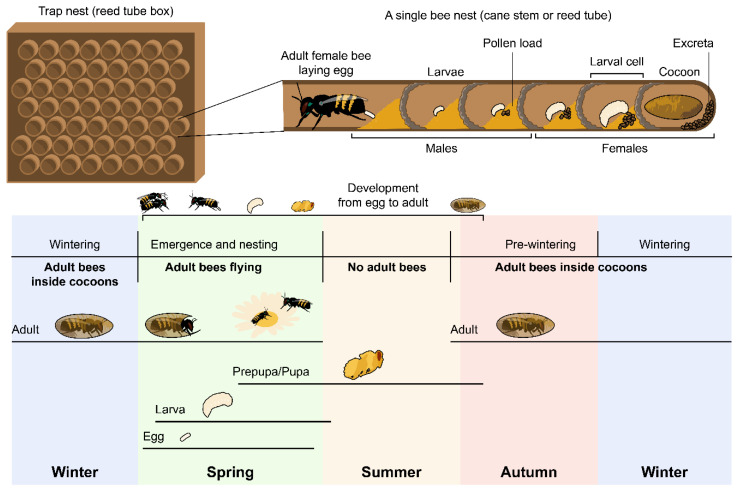
Nesting biology of solitary *Osmia* bees.

**Figure 2 biology-09-00462-f002:**
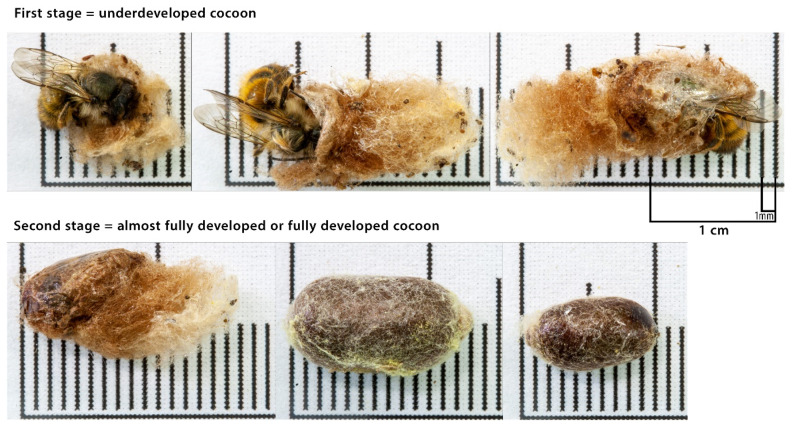
Cocoon stages. Please refer to Section 2.5. *Data Handling and Statistical Analysis* for the description of the cocoon stages.

**Figure 3 biology-09-00462-f003:**
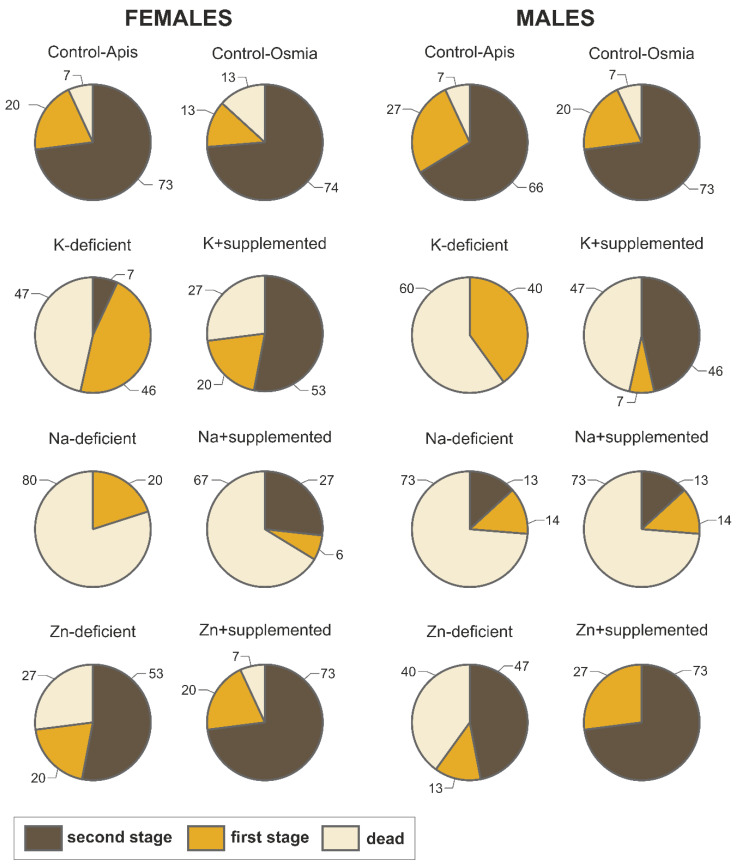
Overall patterns of cocoon developmental stages and mortality (percentage) of *Osmia bicornis* bees (females and males) reared on control pollen pools (Osmia and Apis); pollen pools with scarce levels of K, Na, and Zn; or pollen pools supplemented with certain elements (K, Na, and Zn) to the levels found in the control (Osmia) pollen pools. Cocoon development was distinguished by stage: first stage for undeveloped cocoons and second stage for almost fully or fully developed cocoons. Considering that mortality has the most important and preliminary effect on bee fitness, dead specimens were also included in the graphic to emphasize the overall survival and development patterns for all of the studied individuals. Therefore, all of the percentages were calculated for N = 15 specimens.

**Figure 4 biology-09-00462-f004:**
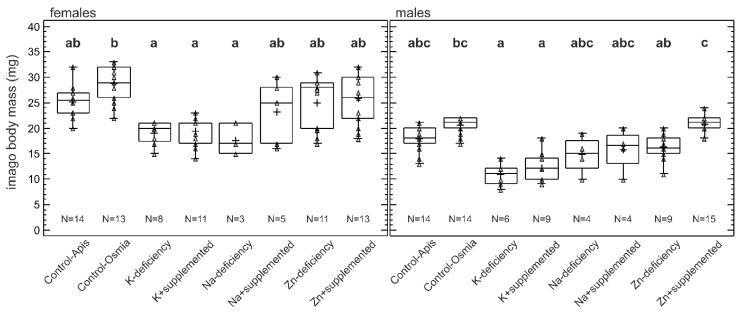
Body masses (mg) of *Osmia bicornis* imagines (females and males) reared on control pollen pools (Osmia and Apis); pollen pools with scarce levels of K, Na, and Zn; or pollen pools supplemented with certain elements (K, Na, and Zn) to the levels found in the control (Osmia) pollen pools. Boxes—lower and upper quartiles, whiskers—minimum and maximum values, plus sign—mean value, centerline—median, and triangles—raw data. (**a**–**c**)—different lowercase letters indicate significant differences between treatments; Kruskal–Wallis (p ≤ 0.05) test with a Bonferroni 95.0% confidence level.

**Figure 5 biology-09-00462-f005:**
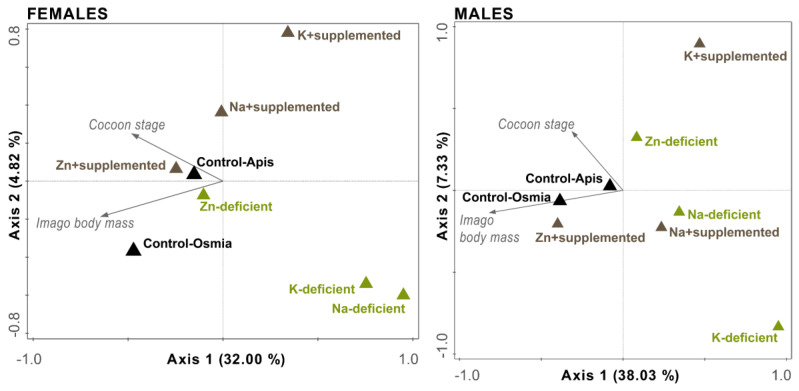
Multivariate analysis of the relationships among the imago body mass, cocoon stage, and experimental diet treatments. The RDA plot and the first two axes are shown. The percentages of variation explained are given for both axes in brackets. Test on all axes: females pseudo-F = 5.8, *p* = 0.002; males pseudo-F = 7.8, *p* = 0.002.

**Table 1 biology-09-00462-t001:** Potassium, sodium, and zinc concentrations (average ± standard deviation; n = 5) measured in pollen pools chosen for the feeding experiment.

Treatment	K		Na		Zn	
	Concentration (ppm)	%	Concentration (ppm)	%	Concentration (ppm)	%
Control-Osmia	6086.2 ± 177.6	100	100.2 ± 47.6	100	57.0 ± 3.3	100
Control-Apis	6144.2 ± 111.0	101	116.4 ± 45.1	116	55.6 ± 2.1	98
K-deficient	4481.2 ± 131.1	74	100.8 ± 27.3	101	53.8 ± 2.6	94
Na-deficient	6127.4 ± 174.2	101	61.4 ± 26.0	61	52.8 ± 3.4	93
Zn-deficient	5914.8 ± 376.1	97	101.0 ± 28.2	101	34.6 ± 1.1	61

The pollen pools corresponded to: (1) control (balanced) diets of *O. bicornis* derived from pollen provisions; (2) a control (balanced) diet derived from commercially available *A. mellifera*-collected pollen; and (3) element-deficient diets sorted from commercially available *A. mellifera*-collected pollen. Percentage—the elemental concentration relative to that in Control-Osmia pollen.

**Table 2 biology-09-00462-t002:** Percentages of mortality and cocoons at the second stage of development in *O. bicornis* female and male bees reared on pollen characterized by different elemental compositions from the 3-day larva to the imago stage. Note that letters and asterisks denote significant differences always within a single sex and between only two treatments (letters: element-deficient vs element + supplemented; asterisks: single treatment vs Control-Apis).

Parameter	Sex	Control-Apis	Control-Osmia	K-Deficient	K+ Supplemented	Na-Deficient	Na+ Supplemented	Zn-Deficient	Zn+ Supplemented
Mortality (%)	Female	7	13	47 * _A_	27 _A_	80 * _A_	67 * _A_	27 _A_	7 _A_
	Male	7	7	60 * _A_	47 _A_	73 * _A_	73 * _A_	40 _A_	0 _B_
2nd cocoon stage (%)	Female	73	74	7 * _A_	53 _B_	0 * _A_	27 * _A_	53 _A_	73 _A_
	Male	66	73	0 * _A_	46 _B_	13 * _A_	13 * _A_	47 _A_	73 _A_

Asterisks denote significant differences in parameters between Control-Apis and the other treatments; _A,B_—different uppercase letters indicate significant differences between the element-deficient and element + supplemented treatments; Chi-squared test with Yates correction (df = 1, *p* ≤ 0.05) performed separately for females and males.

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
