# Peer review of "The Scarcity of Specific Nutrients in Wild Bee Larval Food Negatively Influences Certain Life History Traits"

_biology, 2020, doi:10.3390/biology9120462_

Round 1

Reviewer 1 Report

In this report the authors tested the impact of Na, K or Zn defficiency in mortality, cocoon development and body size of the solitary bee Osmia bicornis. The experiment design included the analyses of 8 different diets, 2 control diets, and six diets defficient or supplemented with any of these nutrients. I found this design quite appropriate to the question proposed. However I found little discussion or comparison with diet control-osmia, which was used as a control for the regime used naturally by this bee specie.

Cocoon develoment was only analysed on bees that reached the imago stage. However, I wondered how was the cocoon in those bees that died during the process. Was the cocoon fully developed? I think this should be discussed.

Body mass was analyzed for all alive individuals after cocoon stage, however, if I understood correctly, there was no differentiation between different cocoon stages. For example, 3 females were analysed in Na-deficiency diet, but there were 0 cocoon developed at second stage. I consider that the correlation between cocoon development and body size should be analysed.

In line 182 it is said that only individual that survived until the end of the experiment were considered to study body mass and cocoons development. So, I wondered if imagos can fully develop in what was classified as “First stage of cocoon”, and if so, if they develop as well as those that reached the “second stage cocoon”? I think this is important as later, in figure 3, we can see that body size was analysed in bees fed the same diet regardless of their cocoon development. Therefore, I think it is important to analyse if there is any correlation between cocoon development and body size as in some diets (Na-deficient female or K-deficient male) no second cocoon stages were reached.

When providing supplemented diets, one would expect a restoration of the phenotype (ie. Mortality, cocoon stage, body mass) observed in Control-apis diet. While this nearly happens in K and Zn supplemented diets, it is far to occur in Na supp diet. This is discussed in the discussion section, where authors agreed that it might be other nutrients in this diet affecting these traits. However, they suggested that addition of Na have a significant effect on female body mass. To me, it looks like that the primary effect of the diet is on the cocoon development, which of course may have an effect of body size, but not as a direct effect. Then, I would suggest again to test the effect of the of cocoon development on body size, rather to analyse all imago together.

It looks like that values of % of cocoons at second and first stage are inverted in diet Zn-supplemented in figure 2.

Correct % of cocoon 2nd stage in female in control-osmia where it corresponds (Table2 or Figure2) for 73 or 74%?

Author Response

Reviewer 1:

Thank you for your valuable comments and for taking the time to review our manuscript.

1. In this report the authors tested the impact of Na, K or Zn defficiency in mortality, cocoon development and body size of the solitary bee Osmia bicornis. The experiment design included the analyses of 8 different diets, 2 control diets, and six diets deficient or supplemented with any of these nutrients. I found this design quite appropriate to the question proposed.

However I found little discussion or comparison with diet control-osmia, which was used as a control for the regime used naturally by this bee species.

Response:

The Control-Osmia diet was used to ensure that the Control-Apis diet itself is indeed a factual control diet, i.e., it does not differ from the Control-Osmia diet in the effects on the bee life history traits studied with regard to elemental composition. In our opinion, since all the treatment diets were performed based on Apis pollen pellets (i.e., Control-Apis), it is biologically and statistically correct to compare these diets with the unmanipulated Control-Apis diet, which has exactly the same characteristics as treatment diets apart from only one factor – specific nutritional manipulation. It is important to emphasize that pollen provisions collected by Osmia sp., and pollen pellets collected by Apis sp. might differ in characteristics other than elemental composition, e.g., microbiota, which can, in turn, affect bee fitness (Dharampal et al., 2020). Thus, to exclude such additional factors, we included a Control-Apis diet with the same elemental composition as the Control-Osmia diet and compared those two. Since there were no differences between Osmia-Control and Osmia-Apis at any of the study endpoints, we proceeded to compare treatment diets with Control-Apis, as such comparisons consider only factual and clear differences caused by specific treatment and nothing else.

2. Cocoon develoment was only analysed on bees that reached the imago stage. However, I wondered how was the cocoon in those bees that died during the process. Was the cocoon fully developed? I think this should be discussed.

Response:

A larva spins its cocoon after its development is completed, undergoes pupation inside the cocoon, and finally emerges into an adult; the bee is wintering in this form. Therefore, if a bee dies during the larval life cycle stage, no cocoon is produced at all. Moreover, we would like to emphasize that in the studied traits and in our discussion, we have focused on only ecologically relevant factors and not on details related to developmental biology. This means that if bees die, their fitness is not related to cocoon development; thus, in our opinion, it makes no ecological sense to discuss the development of cocoons produced by bees that have not reached the imago stage. For clarity, we prefer to focus on the ecological importance of the studied factors. We elucidated these issues in the current version of the manuscript on lines 127-131, 142-144 and 214-223 as well as in the Figure 3 legend. We also added Figure 1 to clarify the life cycle of the bee.

3. Body mass was analyzed for all alive individuals after cocoon stage, however, if I understood correctly, there was no differentiation between different cocoon stages. For example, 3 females were analysed in Na-deficiency diet, but there were 0 cocoon developed at second stage. I consider that the correlation between cocoon development and body size should be analysed.

In line 182 it is said that only individual that survived until the end of the experiment were considered to study body mass and cocoons development. So, I wondered if imagos can fully develop in what was classified as “First stage of cocoon”, and if so, if they develop as well as those that reached the “second stage cocoon”? I think this is important as later, in figure 3, we can see that body size was analysed in bees fed the same diet regardless of their cocoon development. Therefore, I think it is important to analyse if there is any correlation between cocoon development and body size as in some diets (Na-deficient female or K-deficient male) no second cocoon stages were reached.

When providing supplemented diets, one would expect a restoration of the phenotype (ie. Mortality, cocoon stage, body mass) observed in Control-apis diet. While this nearly happens in K and Zn supplemented diets, it is far to occur in Na supp diet. This is discussed in the discussion section, where authors agreed that it might be other nutrients in this diet affecting these traits. However, they suggested that addition of Na have a significant effect on female body mass. To me, it looks like that the primary effect of the diet is on the cocoon development, which of course may have an effect of body size, but not as a direct effect. Then, I would suggest again to test the effect of the of cocoon development on body size, rather to analyse all imago together.

Response:

With all due respect to the reviewer, we are afraid that we do not fully understand this comment. From what we understand, the question is whether bees assigned to the ‘first cocoon development stage’ develop (i.e., reach the same body mass) to the same extent as those assigned to the ‘second cocoon development stage’, which would potentially affect bee fitness. The reviewer asks if the cocoon stage affects the body mass.

We are aware that we may have failed to explain this correctly; however, we treated all the studied traits, i.e., mortality, cocoon stage, and body mass separately because they represent three different sinks into which available sources can be invested. For example, if under optimal conditions a bee has 100% Na to invest and it needs to invest 50% Na to survive, 30% for body mass and 20% for the cocoon, these three traits exert no negative effects. However, if under limited conditions a bee has only 80% of the Na needed, 50% might be invested in survival and 30% in body mass, leaving nothing left for the cocoon and resulting in an undeveloped cocoon in comparison to that developed under optimal conditions. We have now clarified this issue on lines 127-131.

However, according to the reviewer’s advice, we performed additional analysis (RDA), which is described on lines 240-243 and 306-318 and shown in Figure 5.

4. It looks like that values of % of cocoons at second and first stage are inverted in diet Zn-supplemented in figure 2.

Response:

We have corrected this mistake.

5. Correct % of cocoon 2nd stage in female in control-osmia where it corresponds (Table2 or Figure2) for 73 or 74%?

Response:

Thank you for pointing out this mistake. 74% is the right value. We have changed the value in Table 2 (73%) to the correct number – 74%.

Kind regards,
The authors

References for revision:

Dharampal, P. S., Hetherington, M. C., & Steffan, S. A. (2020). Microbes make the meal: oligolectic bees require microbes within their host pollen to thrive. Ecological Entomology, 45(6), 1418–1427. https://doi.org/10.1111/een.12926

Reviewer 2 Report

The authors present a study where they experimentally tested Na, K, and Zn deficiencies in bee-collected pollen on development in Osmia bicornis males and females. The feeding treatments and controls were well designed. Generally, I feel as though the text of the manuscript is acceptable for publication as is. There are some areas of confusion with the results section and how the data are presented, which need to be addressed. 

Specifically, the mortality section (lines 215-228) is very difficult to follow. It would be very helpful to include a figure to visualize statistical differences instead of Table 2. I had a hard time understanding what the A/B significance letters meant in the table, and think a visual representation of the data would be beneficial for the reader. 

Throughout the manuscript, I found the paragraphs to be very long - I would recommend the authors edit the text to break these up into smaller paragraphs. 

Line 312: are the authors referring to Acute Bee Paralysis Virus and Chronic Paralysis Virus? If so, please add "virus" after each.

Something I find very interesting from looking at Figure 2 is that mortality seems to be significantly higher within an ion comparison for the Apis control pollen in both sexes. This indicates to me innate foraging differences between honey bees and O. bicornis which could interact with nutrient to compound the effects of nutrient deficiency. The authors sort of address this on lines 280-296. I think it is less likely this is due to potentially poisonous substances in the pollen and is more likely linked to amino acid differences in the pollen sources. I think this portion of the discussion section could be strengthened. 

Author Response

Reviewer 2:

Thank you for your valuable comments and for taking the time to review our manuscript.

1. Specifically, the mortality section (lines 215-228) is very difficult to follow. It would be very helpful to include a figure to visualize statistical differences instead of Table 2. I had a hard time understanding what the A/B significance letters meant in the table, and think a visual representation of the data would be beneficial for the reader.

Response:

We agree, and we decided to reformat Table 2 and clarified its legend to make it more readable. We have tried to include the results of the data analysis in Figure 2; however, after this modification, the overall appearance of the figure changed to a much less readable and unclear appearance. Therefore, we would prefer to keep Table 2 but in a changed and clarified form.

2. Throughout the manuscript, I found the paragraphs to be very long - I would recommend the authors edit the text to break these up into smaller paragraphs.

Response:

We have edited the text accordingly.

3. Line 312: are the authors referring to Acute Bee Paralysis Virus and Chronic Paralysis Virus? If so, please add "virus" after each.

Response:

No, we are not referring to the virus in this paragraph but rather only to the acute paralysis caused by dysregulation of the K:Na ratio. This has been clarified in the text (lines 366-367).

4. Something I find very interesting from looking at Figure 2 is that mortality seems to be significantly higher within an ion comparison for the Apis control pollen in both sexes. This indicates to me innate foraging differences between honey bees and O. bicornis which could interact with nutrient to compound the effects of nutrient deficiency. The authors sort of address this on lines 280-296. I think it is less likely this is due to potentially poisonous substances in the pollen and is more likely linked to amino acid differences in the pollen sources. I think this portion of the discussion section could be strengthened.

Response:

With all due respect to the reviewer, we do not agree with this comment, as the mortality results presented in Figure 2 are also presented in Table 2, with the outcome of the statistical analysis. According to the statistical analysis (please see more details in the Data Handling and Statistical Analysis section), there were no differences in mortality between the Control-Apis and Control-Osmia treatments for either males or females.

The sentence to which the reviewer is referring states precisely: “A possible explanation is the scarcity of other (than Na) colimiting nutrients or the presence of poisonous substances;(…) ” (lines 342-344 in the current version of the manuscript). Therefore, we clearly state that the scarcity of any limiting nutrient (this of course includes amino acids) might explain the observed effect. We believe that this explanation is sufficient.

Kind regards,
The authors

Reviewer 3 Report

Interesting exploration into nutrient cycling effects on pollinator health. Minor grammatical and formatting suggestions. Qualitative analysis of cocoon development is not sufficiently described to be a new technique, or appropriately cited as a known technique. This gap is increased when reviewing figure 1. Similarly, in table 2, data on cocoon development are presented in a way that makes them appear to be reciprocal, mortality vs cocoons (or mortality + cocoon 2nd stage = 100%). As currently described the relationship between these data, and the total number in each category are not clear. The relationship between available elements manipulated in experimental diet and the development of the cocoons are not clearly defined. Overall the rationale for qualitative analysis based on the development of cocoon rather than metrics of the obtect pupae, adult morphometrics or chemical analysis of cocoon material is unclear. As presented the qualitative analysis of cocoons do not align well with the quantitative components of the experimental design.  Please see comments below:

Line 52-54; review use of "provision" single vs plural, noun vs adjective. Line 54 should at least read as "provisions" for subject verb agreement.

Line 74; needs comma after "therefore"

Line 138; should read "vacuum dryer"

Figure 1.; images presented as "First stage" do not align with description from lines 184-189. In viewing these images the reader (who is likely familiar with the process of metamorphosis) can see fully formed adult bees and disturbed cocoons. As presented there are no obvious deformations visible in adult bees, and condition of the cocoon could easily be interpreted as an artifact of the pupal eclosion process. Does Osmia bicornis consume the inner layer of the cocoon? Is the disturbance of the first stage cocoon from the investigator or from the bee? If all individuals used in this analysis are imago in form, then would the role of the cocoon have already been complete? Additionally, the use of scale, the contrasting cocoon examples, and the adult bees do not help to clarify the methodology. The figure caption does not help the reader to navigate or understand the figure.

Table 1; Consider vertically centering the "treatment" heading

Table 2; Consider vertically centering all one line headings

Section 3.3; currently the authors do not adequately describe the qualitative analysis enough to show how these categories might be analyzed.

Author Response

Reviewer 3:

Thank you for your valuable comments and for taking the time to review our manuscript.

1. Minor grammatical and formatting suggestions.

Response:

The manuscript was edited by a company specializing in the language editing of scientific papers (American Journal Experts, aje.com).

2. Qualitative analysis of cocoon development is not sufficiently described to be a new technique, or appropriately cited as a known technique. This gap is increased when reviewing figure 1. Similarly, in table 2, data on cocoon development are presented in a way that makes them appear to be reciprocal, mortality vs cocoons (or mortality + cocoon 2nd stage = 100%). As currently described the relationship between these data, and the total number in each category are not clear. The relationship between available elements manipulated in experimental diet and the development of the cocoons are not clearly defined. Overall the rationale for qualitative analysis based on the development of cocoon rather than metrics of the obtect pupae, adult morphometrics or chemical analysis of cocoon material is unclear. As presented the qualitative analysis of cocoons do not align well with the quantitative components of the experimental design.

Response:

We thank reviewer for this comment. We agree that the analysis of cocoon development was presented in an ambiguous way. We have clarified these issues. Details are provided on lines 199-205 and 214-223.

3. Line 52-54; review use of "provision" single vs plural, noun vs adjective. Line 54 should at least read as "provisions" for subject verb agreement.

Response:

All linguistic/grammatical issues were checked by American Journal Experts (aje.com) and corrected if needed.

4. Line 74; needs comma after "therefore"

Response:

All linguistic/grammatical issues were checked by American Journal Experts (aje.com) and corrected if needed.

5. Line 138; should read "vacuum dryer"

Response:

This mistake has been corrected.

6. (A) Figure 1.; images presented as "First stage" do not align with description from lines 184-189.

(B) In viewing these images the reader (who is likely familiar with the process of metamorphosis) can see fully formed adult bees and disturbed cocoons. As presented there are no obvious deformations visible in adult bees, and condition of the cocoon could easily be interpreted as an artifact of the pupal eclosion process. Does Osmia bicornis consume the inner layer of the cocoon? Is the disturbance of the first stage cocoon from the investigator or from the bee? If all individuals used in this analysis are imago in form, then would the role of the cocoon have already been complete?

(C) Additionally, the use of scale, the contrasting cocoon examples, and the adult bees do not help to clarify the methodology. The figure caption does not help the reader to navigate or understand the figure.

Response:

(A) We agree that the description was too laconic. We have clarified the description of cocoon stages. Now the text reads as follows: “For adult (imago) bees, the degree of cocoon development was assessed by qualitative analysis, and two stages of cocoon development were distinguished. (1) The first stage was an underdeveloped cocoon that covered only part of the bee or not at all, and the cocoon consisted almost exclusively of soft (“wooly”) fragments; the cocoon tore easily with bare hands but was impossible to cut using a knife because it was too soft. (2) The second stage was an almost fully or fully developed cocoon that covered the whole bee body and mainly consisted of a hard material; the cocoon was difficult or impossible to tear by bare hands but was possible to cut using a knife because it was sufficiently hard (Figure 2).” lines 199-205.

(B) In this qualitative analysis, we considered only cocoon development and not bee development. The experiment lasted for 3 months (early spring to early summer), during which larvae should develop into pupae (pupation) and the pupae should develop into imagines (emergence). A three-month period was chosen to ensure that the bees would pass all the stages until they reached adulthood before the wintering season. The following information is included in the text: “The exposure period was long enough to ensure that all larvae had gone through the life cycle to reach adulthood, i.e., to the stage where fully developed individuals hibernated in their cocoons ” (lines 140-142). We would like to emphasize that adult bees leave the cocoon the next spring, i.e., after another 10 months from pupation and not immediately after they reach the imago stage. Therefore, the cocoons are kept intact by the bees until the next season (Danforth et al., 2019). Moreover, bees do not eat the cocoons at any stage of development, and they leave the cocoon by making an incision with their mandible, not by chewing or nibbling. Thus, the ‘disturbance’ indicated by the reviewer was due to cocoon underdevelopment and not to our manipulation or bee grazing.

Regarding the role of cocoons and their importance for adult bees, we have clarified this issue in the Data Handling and Statistical Analysis section as follows: “The second stage of cocoon development might have the greatest probability of allowing an adult individual to overwinter until the next season and of protecting the bee from external factors (e.g., parasites or pathogens).” (lines 205-207). Considering that bees are exposed to unfavorable conditions during wintering, the cocoon fulfills its role as a buffer against drastic changes due to changes in temperature and humidity as well as blockage against parasites or pathogens.

For additional clarification, we have added Figure 1 showing the nesting and developmental biology of the bee.

(C) The presented photographs show precisely how the cocoons looked like after the experiments were finished (stage 1 cocoons had a bee outside, and stage 2 cocoons had a bee inside). The scale and adult bees were included for reference and to enable the comparison between the presented examples. We believe that without the scale and bee bodies, it would be even more difficult to illustrate the degree of development of the cocoon given that one criterion is that the cocoon should cover the body of the insect. The chosen examples were included to cover all the possibilities of deformation/development of cocoons that were found during the evaluation.

7. Table 1: Consider vertically centering the "treatment" heading

Response:

The tables as well as the whole text were formatted by American Journal Experts (aje.com) according to the journal’s guidelines.

8. Table 2; Consider vertically centering all one line headings

Response:

The tables as well as the whole text were formatted by American Journal Experts (aje.com) according to the journal’s guidelines.

9. Section 3.3; currently the authors do not adequately describe the qualitative analysis enough to show how these categories might be analyzed.

Response: We have clarified this issue in the response to the second question.

Kind regards,
The authors

References for revision:

Danforth, B. N., Minckley, R. L., & Neff, J. L. (2019). The Solitary Bees: Biology, Evolution, Conservation. Princeton University Press. https://press.princeton.edu/books/hardcover/9780691168982/the-solitary-bees